# Cryopreservation and Maturation Media Optimization for Enhanced Somatic Embryogenesis in Masson Pine (*Pinus massoniana*)

**DOI:** 10.3390/plants14111569

**Published:** 2025-05-22

**Authors:** Qian Yang, Ying Lin, You-Mei Chen, Qi Fei, Jian-Ren Ye, Li-Hua Zhu

**Affiliations:** 1Collaborative Innovation Center of Sustainable Forestry in Southern China, Nanjing 210037, China; 3220100088@njfu.edu.cn (Q.Y.); linin77@njfu.edu.cn (Y.L.); chenyoumei@njfu.edu.cn (Y.-M.C.); feiqi1999@njfu.edu.cn (Q.F.); jrye@njfu.edu.cn (J.-R.Y.); 2Institute of Forest Protection, College of Forestry and Grassland, Nanjing Forestry University, Nanjing 210037, China

**Keywords:** *Pinus massoniana*, somatic embryogenesis, cryopreservation, maturation, transplanting

## Abstract

*Pinus massoniana* Lamb. (masson pine) is a critical species for afforestation in southern China but faces severe threats from pine wilt disease (PWD) caused by *Bursaphelenchus xylophilus*. To accelerate disease-resistant breeding, this study investigated the effects of cryopreservation on the embryonic capacity of the embryogenic callus as well as the effects of abscisic acid (ABA), polyethylene glycol 8000 (PEG 8000) and phytagel concentration on the somatic embryo’s maturation and germination. Furthermore, the impact of transplanting substrates on the survival and growth of regenerated plantlets were evaluated. The results showed that cryopreservation at −196 °C effectively maintained the embryogenic potential of the callus, with post-thaw tissues exhibiting superior somatic embryo maturation capacity compared to the long-term subcultured callus (38.4 vs. 13.2 embryos/mL). Key maturation parameters were systematically optimized: ABA concentration at 6 mg/L in the suspension culture maximized embryo yield of 24.1 somatic embryos/mL, while PEG 8000 at 130 g/L in solid medium achieved peak embryo production of 38.4 somatic embryos/mL, and the maximum of 26.6 somatic embryos/mL when the concentration of phytagel was 3.5 g/L. The highest germination rate of 29.8% was observed with 130 g/L PEG in the maturation medium. The highest survival rate (56.5%) and maximum plant height (22.3 cm) after 12 months of transplantation were achieved in substrates consisting of soil and vermiculite, which outperformed those containing varying proportions of mushroom residue. This study establishes a scalable protocol for the mass propagation of PWD-resistant *P. massoniana*, integrating cryopreservation and maturation media optimization, which offers dual benefits for disease-resistant breeding and sustainable germplasm conservation.

## 1. Introduction

*Pinus massoniana* Lamb. (masson pine) is one of the most important native tree species for afforestation in southern China. It is widely planted in China due to its significant economic value and ecological service. Additionally, masson pine is renowned for its rapid growth and tolerance to drought and arid conditions [1]. However, the species faces a severe threat from pine wilt disease (PWD), which is caused by the pine wood nematode (PWN, *Bursaphelenchus xylophilus*). PWD has been reported in several countries, including Japan (first detected in 1905), China (1982), Korea (1988), Portugal (1999), and Spain (2008) [2,3,4]. Most recently, it was detected in the Republic of Armenia in 2022–2023 [5]. Despite comprehensive management strategies, including strict quarantine protocols, sanitation logging, biological control agents, and chemical interventions, the spread of PWD remains a significant challenge [6], constituting a significant biotic threat to global *Pinus* ecosystems [7]. In the face of such a destructive disease, resistance breeding has emerged as a critical long-term strategy to enhance the natural defenses of pine trees against PWN. The first resistance breeding program against PWN was initiated in Japan in 1978, focusing on selecting resistant Japanese black pines (*Pinus thunbergii*) and Japanese red pines (*Pinus densiflora*) [8]. In China, resistance breeding of *P. massoniana* to PWD was initiated in Anhui Province, in 2001 [9,10]. However, traditional breeding strategies are hindered by the long reproductive cycles and high heterozygosity of conifers, limiting their efficiency in developing PWD-resistant genotypes [11,12].

Somatic embryogenesis (SE) is a promising tool for rapid clonal propagation of elite conifers, offering advantages such as fast multiplication and large-scale production of genetically uniform, disease-resistant plantlets [13]. Since 1985, when the SE from immature zygotic embryos of *Picea abies* [14] and *Larix decidua* [15] was reported, in vitro embryogenesis of conifers has made remarkable progress. SE technology has been successfully applied to at least 36 *Pinus* species, including *P. massoniana* [16], *P. koraiensis* [17], *P. radiata* [18], *P. thunbergia* [19], and *P. elliottii* [20], and so on. Despite these advances, maintaining the embryogenic potential of the embryogenic callus (EC) remains a critical challenge for the success of SE. The embryogenic potential of ECs is gradually lost with repeated subculturing over time, significantly impacting somatic embryo formation [21]. This issue not only limits the application of SE technology in conifers but also complicates the selection of superior genotypes.

Cryopreservation, particularly using liquid nitrogen (−196 °C), has emerged as a most reliable and cost-effective solution to preserve embryogenic competence [22]. Studies in *P. koraiensis* and *P. pinaster* demonstrate that cryopreserved ECs retain high embryogenic potential and produce viable somatic embryos and regenerated plants [23,24]. Notably, in *Abies nordmanniana*, stepwise temperature changes during freezing and thawing, combined with abscisic acid (ABA) as a cryoprotectant, significantly improved post-thaw survival of embryogenic callus and subsequent somatic embryo development [25]. This approach minimized membrane lipid peroxidation and maintained genetic stability, critical for long-term storage in gene banks [26]. For instance, in *P. abies*, slow cooling protocols using a Mr Frosty vessel (1 °C/min) enhanced callus vitrification and reduced ice crystal formation, thereby preserving totipotency [27]. Concurrently, maturation media formulations, particularly the interplay of abscisic acid (ABA), polyethylene glycol 8000 (PEG 8000), phytagel, activated carbon (AC) and maltose critically influence embryo maturation and germination [28,29,30]. Yet, the synergistic effects of these factors in cryopreserved ECs are poorly understood. Acclimatization stage also remains a major challenge in conifer micropropagation, with substrate conditions during transplantation critically affecting plant survival and growth [31].

Building on these gaps, this study aims to optimize the SE protocol for *P. massoniana* by focusing on the following objectives: (1) Evaluating cryopreservation’s efficacy in maintaining embryogenic potential and post-thaw SE capacity. (2) Optimizing maturation media through ABA, PEG 8000, and phytagel concentration gradients. (3) Assessing transplanting substrates to maximize the survival and growth of the regenerated plantlet.

## 2. Results

### 2.1. Cryopreservation of Embryogenic Callus of P. massoniana

#### 2.1.1. Post-Thaw Recovery Rate of Embryogenic Callus in *P. massoniana*

In this study, five cryopreserved ECs of the GX 1-3-5 cell line of *P. massoniana* were thawed, and cultured for two weeks under standard proliferation conditions. After this period, only one of the five samples exhibited viable proliferation capacity, resulting in a post-thaw recovery rate of 20%. The recovered cell line was designated as JD 1-3-5.

#### 2.1.2. Microstructure of *P. massoniana* Embryogenic Callus Before and After Cryopreservation

Double-staining observations of the cell line GX1-3-5 before and after cryopreservation, as well as long-term culture for 10 months, revealed significant differences in embryogenic characteristics. Pre-cryopreservation cultures exhibited characteristic proembryogenic masses (PEM) III and SE stages, with intact embryonic head and suspensor cells (Figure 1D). In contrast, the cells cultured on proliferation medium for 10 months showed split embryonic head cells, unaggregated suspensor cells, and the cell mass was dispersed, indicating a loss of embryonic properties (Figure 1F). Notably, the cells that resumed growth after thawing were predominantly in the SE stage, exhibiting distinct bundle-like head and suspensor structures, with aggregated embryonic head cells and clustered suspensor cells, suggesting that cryopreservation preserved their embryogenic capacity (Figure 1E). In summary, the cell line GX1-3-5 lost its embryonic properties due to long-term subculture, while it retained embryonic structures after 6 months of cryopreservation.

### 2.2. Maturation of Somatic Embryos of P. massoniana

#### 2.2.1. Effect of ABA Concentrations on Somatic Embryo Maturation

The double staining assay revealed that varying ABA concentration (0–10 mg/L) in the suspension solution has a limited impact on the developmental stages of GX1-3-5, during which cells mostly stayed at the SE stage and a minority at the PEM III stage (Figure 2A). After 75 days cultured on solid maturation medium, somatic embryos developed on medium supplemented with 2 mg/L ABA exhibited normal morphology (Figure 2B,D) compared to those on ABA-free media (Figure 2C,E). ABA concentrations of 0–10 mg/L had significant effects on the yield of somatic embryos (*p* < 0.05). Pretreatment with 6 mg/L ABA in suspension culture, followed by maturation on 2 mg/L ABA solid medium yielded peak embryo production (24.1 ± 14.6 embryos/mL, Figure 2F), whereas maturation on ABA-free medium resulted in lower yields (9.8 ± 6.7 embryos/mL, Figure 2G) with predominantly abnormal embryos. Both higher (> 6 mg/L) and lower (< 4 mg/L) ABA pretreatment concentrations induced callus browning. These findings suggest that the presence of 2 mg/L ABA in solid maturation medium is essential for normal somatic embryo morphology, whereas a 6 mg/L ABA pretreatment in suspension culture optimizes embryo yield.

#### 2.2.2. Effect of PEG 8000 Concentrations on Somatic Embryo Maturation

After 75 days of culture on maturation medium containing different concentrations of PEG, distinct differences were observed in somatic embryo development between the thawed cryopreserved callus and long-term subcultured callus. Embryos from thawed cryopreserved callus exhibited normal development, showing differentiation into normally developed somatic embryos (Figure 3A). In contrast, embryos from long-term subcultured callus were predominantly columnar (embryos that are elongated and cylindrical in form, lacking the typical differentiation into distinct embryonic structures such as cotyledons), with few exhibiting normal development (Figure 3B).

PEG concentrations of 130–190 g/L had significant effects on the yield of somatic embryos (*p* < 0.05). As PEG concentration increased, the number of somatic embryos produced by thawed cryopreserved callus decreased. The highest number of somatic embryos was observed at 130 g/L PEG (38.4 ± 17.2/mL), but these embryos were mostly columnar and undeveloped compared to those formed at 150 g/L PEG (Figure 3A). No significant difference in the number of somatic embryos was found between PEG concentrations ranging from 150 to 190 g/L.

For the long-term subcultured callus, the number of somatic embryos first increased and then decreased with increasing PEG concentrations. The highest yield was 13.2 ± 5.5 embryos/mL at 170 g/L PEG, although these embryos were mostly columnar (Figure 3B). Overall, somatic embryos from thawed cryopreserved callus were superior in both quality and quantity to those from the long-term subcultured callus.

#### 2.2.3. Effect of Phytagel Concentrations on Somatic Embryo Maturation

After 75 days of culture on maturation medium containing varying concentrations of phytagel, it was observed that the thawed callus successfully differentiated into somatic embryos (Figure 4A). With the phytagel concentration in the maturation medium increased, the number of somatic embryos first increased and then decreased. The highest number of somatic embryos (26.6 ± 7.0/mL) was produced when the concentration of phytagel was 3.5 g/L (Figure 4C). While somatic embryo production decreased at phytagel concentrations of 4.5 and 5.5 g/L, embryo development remained normal. At lower phytagel concentrations of 1.5 and 2.5 g/L, the callus exhibited severe waterlogging (characterized by a water-soaked appearance and a translucent texture), which hindered embryo development (Figure 4B).

### 2.3. Somatic Embryo Germination and Plantlet Regeneration

After approximately 75 days on the maturation medium, somatic embryos were transferred to the germination medium. Within 1–2 months, these embryos elongated, and their cotyledons gradually unfolded. Somatic embryos derived from thawed cryopreserved callus, treated with different PEG concentrations, successfully germinated and grew into regenerated plantlets (Figure 5A). In contrast, somatic embryos from long-term subcultured callus showed browning, stunted growth, and mostly failed to germinate (Figure 5B). Maturation treatments with varying PEG concentrations had no significant effect on somatic embryo germination; when the PEG concentration was 130 g/L, the highest germination rate of somatic embryos from JD 1-3-5 was 29.8 ± 5.2% (Figure 5C), while the highest germination rate of somatic embryos from long-term subculture callus was 14.5 ± 16.9% (Figure 5C). These results indicate that the callus of long-term subculture lost its embryogenic potential, whereas cryopreservation effectively maintains the callus’s capacity for somatic embryo maturation and germination.

### 2.4. Acclimatization and Transplantation of Regenerated Plantlets

For rooting induction, somatic embryos were transferred to rooting medium. After 2–3 months, the regenerated plantlets were transferred from sterile tissue-culture containers to plastic pots with various transplant substrates for acclimatization. After 3 months, it was observed that different transplant substrates had a significant impact on plant height growth (*p* < 0.05). The addition of mushroom residue to the substrate enhanced plant growth and survival rates, with plantlets grown in substrates without mushroom residue having a survival rate of only 56.5% (Table 1). However, after 12 months, the results were completely different. In substrates without mushroom residue, the survival rate of regenerated plantlets was still 56.5%, and their plant height reached 22.3 cm. In contrast, plantlets grown in substrates with mushroom residue exhibited poor growth, and their survival rate dropped below 30% (Table 1, Figure 6). The results indicate that adding mushroom residue to the substrate may be unfavorable for the long-term growth and survival of regenerated plantlets.

## 3. Discussion

Repeated subculturing of ECs leads to a progressive decline in embryogenic potential, which is a major bottleneck for large-scale SE in conifers. In breeding programs, SE and cryopreservation are essential for the propagation of superior genotypes, especially considering that the differentiation potential of ECs has been shown to decline over time [32,33,34]. In this context, the successful cryopreservation of ECs plays a critical role in maintaining regenerative capacity and preserving valuable germplasm [22]. Our findings reinforce this concept by demonstrating that cryopreserved ECs retained significantly higher maturation and germination capacities compared to long-term subcultured lines. This result is consistent with prior reports in *P. koraiensis* [35] and *P. pinaster* [23]. Genotype is a critical factor influencing somatic embryogenesis. In future studies, we will employ diverse genotypes to conduct relevant experiments, thereby ensuring the validity of the methods utilized and the results obtained in this paper.

Microscopic examination of cryopreserved and long-term subcultured ECs revealed notable differences. Cryopreserved ECs retained intact embryonic heads and well-organized suspensors, while long-term subcultured cells exhibited split heads and non-converging suspensor structures. These observations indicate that cryopreservation is effective in preserving the morphological stability of *P. massoniana* embryogenic callus, consistent with Shen et al. and Lineros et al. This structural preservation is crucial for maintaining totipotency, as disorganization of embryonic domains often correlates with reduced maturation capacity. Similar results were observed in *P. radiata*, where PEMs maintained their structural integrity after cryopreservation, showing no visible cellular damage under optical microscopy [36,37]. Hazubska-Przybył et al. observed similar results in *A. alba* × *A. numidica* and *Pinus nigra*, where cryopreserved tissues retained viable embryogenic regions, despite suspensor destruction, and were successfully regenerated post thaw [38].

Cryopreservation is highly influenced by the cryoprotectants and freezing methods. In this study, we utilized a cryoprotectant combination consisting of 5% DMSO, 80% sucrose (0.5 mol/L), and 15% PEG 4000, which effectively preserved the embryogenic potential of *P. massoniana* ECs, as shown by the significantly higher maturation and germination capacities observed in cryopreserved ECs compared to long-term subcultured lines. These findings are consistent with those reported by Gao et al. for *P. koraiensis*. The suboptimal post-thaw recovery rate of callus tissues may stem from the sensitivity of embryogenic callus to cryopreservation stress, cellular damage during freezing and thawing, and the need to further optimize the cryoprotectant mixture and cooling protocol. Although DMSO is commonly used in conifer cryopreservation for its ability to prevent ice crystal formation, its cytotoxic effects, including DNA damage and apoptosis, have been well-documented and may negatively impact somatic embryo development post thaw [36,38]. To mitigate these effects, alternative cryoprotectants such as glycerol, sucrose, and proline have been explored, which help reduce cellular damage and improve survival and regeneration rates of cryopreserved ECs, as shown in studies on *Abies* and *Pinus* species [24,39]. In our future research, we plan to explore the use of these alternative cryoprotectants in cryopreservation studies. Assessing genetic stability during the somatic embryogenesis process is crucial for large-scale production. The decline in the regenerative capacity of embryogenic tissues is often associated with genetic instability. In our study, we evaluated the regenerative potential of the embryogenic tissues post thaw but did not assess their genetic stability. Future research will focus on this important aspect.

Most studies on SE in conifers have focused on optimizing culture conditions for the maturation stage [40,41]. Somatic embryo maturation is influenced by several factors, including genotype, ABA, gelling agent, and sucrose concentration in the maturation medium [42,43,44,45]. ABA has been confirmed to play a crucial role in regulating somatic embryo formation and development [46,47]. In our study, ABA was added to the liquid medium prior to transferring the callus to the solid maturation medium to regulate the state of the ECs. Significant differences in somatic embryo production were observed after pretreatment with different concentrations of ABA. The highest yield (24.1 ± 14.6 somatic embryos per mL) was achieved with 6 mg/L ABA, while the production was minimal without ABA. This indicates that adding an appropriate concentration of ABA during the liquid suspension culture stage enhances somatic embryo maturation in *P. massoniana*. Yang et al. found that adding 5 mg/L ABA to the liquid medium significantly increased somatic embryo production in *P. elliottii* [20]. Li et al. reported that suspending ECs in liquid medium containing 37.84 μM ABA resulted in up to 274 somatic embryos per mL in *P. elliottii* × *P. caribaea* [48]. These results confirm that ABA promotes somatic embryo maturation in various pine species [49]. Moreover, the response of somatic embryos to exogenous ABA is genotype-dependent in conifers [46,50,51].

Maintaining high and stable osmotic pressure is critical for generating high-quality somatic embryos. PEG can induce water stress that simulates arid conditions, reduces cellular water content and enhances embryo maturation [52]. In our study, the highest number of somatic embryos (38.4/mL) was observed at a PEG concentration of 130 g/L, with a decline in embryo production as the concentration increased further. Similarly, Li et al. found that somatic embryo production peaked at 80 g/L PEG, with significantly reduced cotyledonary embryo formation beyond this threshold [48]. This indicates that adding an optimal PEG concentration in the maturation medium effectively enhances somatic embryo maturation. The type and concentration of gelling agents can influence SE by regulating the osmotic pressure of the medium, thereby altering water balance and nutrient uptake. Phytagel, a commonly used gelling agent, provides structural support to explants and influences the medium’s water potential. High concentrations of phytagel may enhance cellular reprogramming through induced water stress. Avila-Victor et al. reported that increasing phytagel concentration from 2.3 to 5.0 g/L doubled the number of somatic embryos obtained [30]. In our study, the highest number of somatic embryos (26.6/mL) was achieved at 3.5 g/L phytagel, with yields decreasing at higher concentrations, demonstrating that phytagel concentration affects somatic embryo formation. Phytagel concentration significantly affects medium osmotic potential. Future studies should investigate the precise phytagel–osmotic pressure relationship to better elucidate its impact on embryo development.

Germination of mature embryos and conversion into somatic seedlings remain major challenges in conifer SE [53]. We investigated the impact of maturation treatments on germination and found that the highest germination rate of somatic embryos (29.8%) was achieved at a PEG concentration of 130 g/L. Chen et al. reported that the germination rate initially increased and then decreased with rising PEG concentration in the maturation medium, peaking at a germination rate of 75.3% when the PEG concentration was 110 g/L [29]. These findings demonstrate that PEG concentration in the maturation medium has a lasting effect on the germination of *P. massoniana* somatic embryos. Our study suggests that a PEG concentration of 130 g/L in the maturation medium is optimal for both somatic embryo maturation and germination.

Plantlets formed under aseptic conditions often exhibit reduced resistance due to the lack of interaction with microorganisms, resulting in lower survival rates after transplantation [54]. The acclimatization stage is a significant challenge in conifer micropropagation [31]. The transplanting substrate significantly influences the growth and development of seedlings, requiring properties such as breathability, water retention, nutrient provision, and anchorage. Hu et al. reported that using mushroom residue compost as a substitute for peat in cultivating *P. tabulaeformis* container seedlings significantly improved biomass and root development compared to peat-based substrate [55]. In this study, without mushroom residue in the substrate, regenerated plantlets achieved a maximum survival rate of 56.5%, with plant height growth reaching 22.3 cm (12 months after transplantation). Conversely, incorporating mushroom residue into the substrate decreased both growth and survival rates, with the survival rate dropping below 30%, indicating no promotional effect from the mushroom residue on plant growth. It is worth noting that the presence of mushroom residue in the substrate may introduce potential mycotoxin contamination, which could have an impact on plant health and growth.

## 4. Materials and Methods

### 4.1. Cryopreservation of Embryogenic Callus of P. massoniana

The cell line GX 1-3-5, previously obtained in the laboratory of Forest Protection at Nanjing Forestry University (Nanjing, Jiangsu Province, China) [29], was cultured on proliferation medium prior to cryopreservation.

After 14 days of proliferation, 0.5 g of embryogenic callus (EC) was transferred to proliferation medium with 0.5 mol/L sucrose for 48 h. Then, the pretreated callus was transferred into 1.8 mL cryotubes, immersed in 1.5 mL cryoprotectant solution which was prepared by mixing 5% (*v*/*v*) dimethyl sulfoxide (DMSO), 80% (*v*/*v*) 0.5 mol/L sucrose, and 15% (*v*/*v*) polyethylene glycol 4000 (PEG 4000) [Note: PEG 4000 solution was prepared by dissolving 250 g PEG 4000 in sterile distilled water to a final volume of 500 mL (50% *w*/*v* stock)], and subjected to programmed cooling (Planer, Kryo 560-16) under the following protocol: equilibration at 0 °C for 10 min, cooling at 1 °C/min to −80 °C, holding at −80 °C for 30 min, then rapidly transfer the cryotubes to liquid nitrogen (−196 °C) for long-term storage.

After 6 months of cryopreservation, the cryotubes were retrieved and thawed in a 32 °C water bath. The callus was rinsed three times with 0.5 mol/L sucrose solution, followed by a final rinse with sterile water. Recovered callus (designated as JD 1-3-5) was transferred to proliferation medium and cultured at 24 ± 1 °C in the dark for two weeks. Callus exhibiting normal proliferation was selected for subsequent SE capacity evaluation.

Unless otherwise stated, the proliferation medium consisted of the LP medium [56] supplemented with: 2,4-dichlorophenoxyacetic acid (2,4-D) 1 mg/L, 6-benzylaminopurine (6-BA) 0.5 mg/L, inositol 1 g/L, L-glutamine 0.45 g/L, maltose 15 g/L, ascorbic acid (vitamin C) 0.25 mg/L, casein hydrolysate 0.5 g/L, and agar 6.24 g/L. The medium pH was adjusted to 5.8 with 1 mol/L KOH and HCl before autoclaving at 121 °C for 21 min.

### 4.2. Microscopic Observation of Embryogenic Callus

First, embryogenic callus was stained with 2% (*w*/*v*) acetocarmine for 30 s and 0.05% (*w*/*v*) Evans blue for 5 s on glass slides, followed by rinsing the dyes off with water. The morphological structure of embryogenic callus was then observed under a Zeiss stereo microscope, SteRo Discovery V20 (Carl Zeiss, Germany). The microstructures of embryogenic callus tissues under different treatments (long-term subculture, before cryopreservation, and after thawing) were observed microscopically three times each.

### 4.3. Maturation of Somatic Embryos of P. massoniana

For the establishment of liquid culture, 1.0 g of fresh embryogenic tissue was transferred from proliferation media into 100 mL Erlenmeyer flasks containing 30 mL of liquid medium. The liquid medium consisted of LP medium with 0.5 mg/L 2,4-D, 0.25 mg/L 6-BA, 15.0 g/L maltose, 1.0 g/L inositol, 0.50 g/L casein hydrolysate and 0.45 g/L glutamine, flasks were incubated on a rotary shaker (90 rpm) in darkness at 25 ± 1 °C. Seven days after the suspension culture was established, the culture was transferred into measuring cylinders, settling for 20 min; the supernatant was then removed. The suspension cells (4.0 mL) were dispersed on sterilized filter paper, which was placed on the surface of the solid maturation medium. The maturation medium contained LP basic medium supplemented with 2.0 mg/L ABA, 0.5mg/L gibberellic acid (GA3), 0.25 mg/L VC, 25 g/L maltose, 130 g/L PEG 8000, 1.0 g/L inositol, 1.0 g/L glutamine, 1.0 g/L AC and 3.5 g/L phytagel. Each treatment was conducted with at least five replicates, each represented by one Petri dish. All cultures were cultured at 24 ± 1 °C in the dark for 75 days, then the number of somatic embryos was recorded.

#### 4.3.1. Effect of Pretreatment with Different Concentrations of ABA on Somatic Embryo Maturation

The cell line GX1-3-5 was used as the material. A suspension culture system was established following the method described above, the suspended cells were cultured with shaking for 7 days. Afterward, the culture was allowed to stand for 20 min, and the supernatant was discarded. The cell suspension was then inoculated into a liquid medium (inoculation ratio of 1:5) containing different concentrations of ABA, with ABA concentrations set at 0, 2, 4, 6, 8, and 10 mg/L. After 7 days of pretreatment, 4.0 mL of the cell suspension was transferred onto the maturation medium containing 2 mg/L ABA. Additionally, a separate solid maturation medium without ABA was prepared to assess the role of ABA in somatic embryo maturation, with all other treatment conditions identical to those described above. Five replicates of each treatment were incubated at 24 ± 1 °C in the dark for 75 days.

#### 4.3.2. Effect of PEG 8000 Concentrations on the Maturation Capacity of Embryonic Callus After Thaw Recovery and Long-Term Culture

The cell line JD 1-3-5 was used as material. A suspension culture system was established following the method described above. Then, 4.0 mL of suspension cells was transferred onto maturation medium, and the concentration of PEG 8000 in the maturation medium was set at 110, 130, 150, 170, and 190 g/L, with five replicates for each treatment, and cultured at 24 ± 1 °C in the dark for 75 days. The somatic embryo production of each treatment was observed and the quantities counted. At the same time, the same maturation test was performed on the cell line GX1-3-5, which had been long-time subcultured for 10 months.

#### 4.3.3. Effect of Phytagel Concentrations on the Maturation of Embryonic Callus After Thaw Recovery

The cell line JD 1-3-5 was used as material. After establishing the suspension culture system as described above, 4.0 mL of suspension cells were transferred onto maturation medium, and the concentration of phytagel in the maturation medium was set at 1.5, 2.5, 3.5, 4.5, 5.5 g/L, with five replicates for each treatment, and cultured at 24 ± 1 °C in the dark for 75 days. The number of somatic embryos were counted.

### 4.4. Somatic Embryo Germination and Plantlet Regeneration

Mature cotyledonary embryos were directly transferred to germination medium consisting of LP medium with 20 g/L maltose, 2.0 g/L AC and 8 g/L agar. Culture plates were incubated for 5–7 days in the dark followed by being placed under a 16 h photoperiod provided by cool white fluorescent tubes and cultured at 24 ± 1 °C. When germinated after 1–2 months, somatic embryos were aseptically transplanted to the rooting medium (WPM) which consisted of 0.2 mg/L a-naphthaleneacetic acid (NAA), 1.0 mg/L indole-3-butyric acid (IBA), 10 g/L sucrose, 0.1 g/L inositol, 0.5 g/L AC and 10 g/L carrageenan. Regenerated plantlets were obtained after about 6–8 weeks of culture in the light.

#### 4.4.1. Effect of Different Maturation Treatments on the Germination of Somatic Embryos

To investigate the effect of cryopreservation on the regeneration ability of *P. massoniana* ECs, mature somatic embryos obtained from cell line JD 1-3-5 and GX 1-3-5 were used as materials for somatic embryo germination.

#### 4.4.2. Effect of Transplanting Substrate on Growth and Survival of Regeneration Plantlets

Regenerated plantlets obtained from cell line JD 1-3-5 were used as materials, the regenerated plantlets with similar growth status were transferred to plastic pots (10 cm in diameter, 8.5 cm in height) filled with different substrates, the ratios of nutrient soil, perlite, and mushroom residue of 3:1:0, 3:1:1, 3:1:2, and 3:1:3, and 23 plantlets, respectively, were transplanted into each substrate. The regenerated plantlets were acclimatized in a growth cabinet, as described in Sun et al. [19]. The survival rate and plant height were recorded after transplantation.

### 4.5. Statistical Analysis

Data were analyzed by a one-way analysis of variance (ANOVA) using SPSS 26.0 and GraphPad Prism Software 8.0. To test the significant differences between the mean values, the Duncan’s multiple range (DMRT) test was performed at *p* < 0.05. Data represent mean ± standard deviation (SD). The data consists of somatic embryo yield, germination rate, plant growth and survival rate.

## 5. Conclusions

This study developed an effective protocol for the mass propagation of *P. massoniana*, integrating cryopreservation, somatic embryo maturation optimization, and transplant substrate selection. Cryopreservation was shown to maintain the embryogenic potential of *P. massoniana* callus, thereby enabling the long-term preservation of valuable cell lines. Optimized maturation conditions, including ABA at 6 mg/L, PEG at 130 g/L, and phytagel at 3.5 g/L, significantly enhanced somatic embryo production and germination rates. Additionally, the highest survival rate (56.5%) and improved plant growth were achieved with a transplanting substrate consisting of nutrient soil and perlite in a 3:1 ratio, without the addition of mushroom residue. These results offer a reliable protocol for the propagation of disease-resistant *P. massoniana*, with broad implications for both breeding programs and the sustainable conservation of germplasm resources. Further validation using multiple genotypes is recommended to enhance the robustness and scalability of the protocol.

## Figures and Tables

**Figure 1 plants-14-01569-f001:**
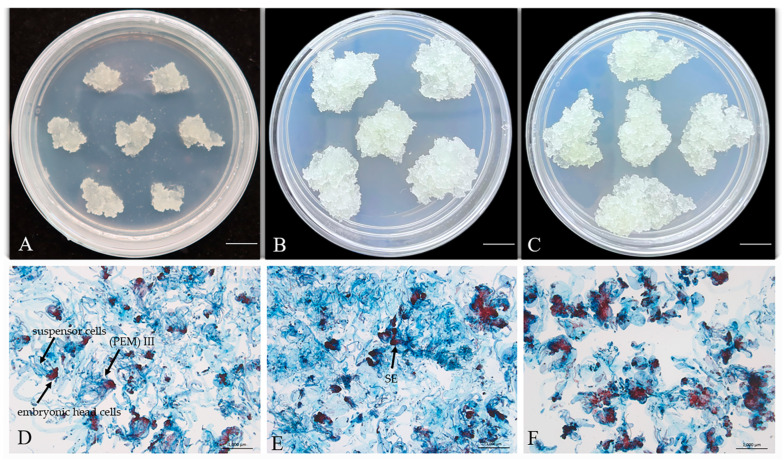
Morphology and microstructure of *P. massoniana* callus before cryopreservation, after cryopreservation, and long-term subculture callus (GX1-3-5). (**A**,**D**) Fresh callus before cryopreservation. (**B**,**E**) Callus recovered after cryopreservation. (**C**,**F**) Callus subjected to long-term subculture. Scale bar: 1.0 cm (**A**–**C**), 1.0 mm (**D**–**F**).

**Figure 2 plants-14-01569-f002:**
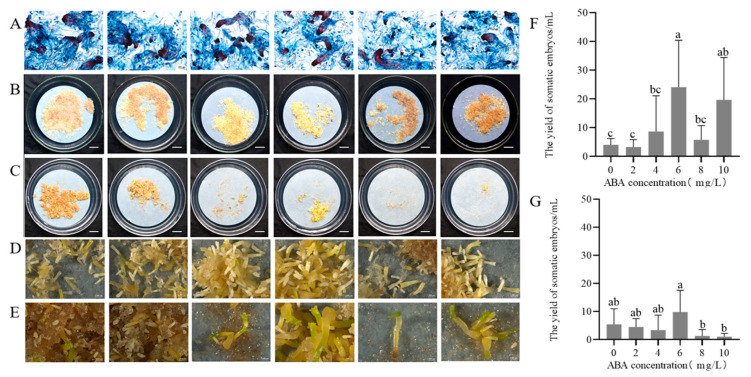
Effect of ABA pretreatment at different concentrations on somatic embryo maturation of *P. massoniana* (75–day culture). (**A**) Microstructure of embryogenic cells under varying ABA pretreatment concentrations. Scale bars = 0.02 cm. (**B**) Somatic embryos matured on medium supplemented with 2 mg/L ABA (pretreatment concentrations of ABA from left to right: 0, 2, 4, 6, 8, and 10 mg/L). Scale bar = 1.0 cm. (**C**) Somatic embryos matured on ABA-free medium (pretreatment concentrations same as in B). Scale bar = 1.0 cm. (**D**,**E**) Stereomicroscope close-up views of (**B**) and (**C**), respectively. Scale bars = 0.1 cm. (**F**,**G**) Yield of somatic embryos derived from varying ABA pretreatments on maturation medium with 2 mg/L ABA (**F**) or without ABA (**G**). Data represent mean ± SD. Different lowercase letters above the bars indicate the differences among ABA treatments within the same embryogenic cell line according to ANOVA and Duncan’s test (*p* < 0.05).

**Figure 3 plants-14-01569-f003:**
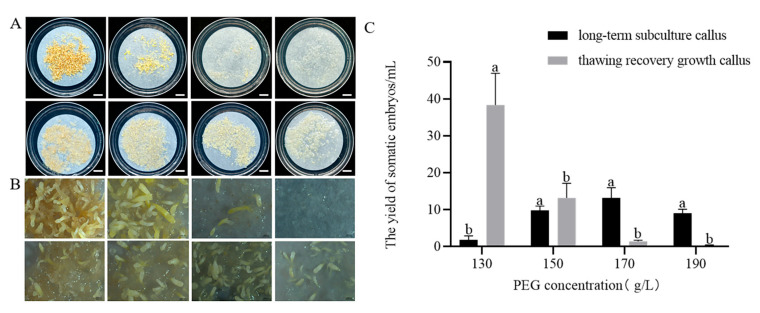
Effect of PEG concentrations on somatic embryo maturation from cryopreserved and long-term subcultured callus in *P. massoniana* (75-day culture). (**A**) Somatic embryos matured on medium containing different PEG concentrations (left to right: 130, 150, 170, and 190 g/L). Upper row: embryos from thawed cryopreserved callus; lower row: embryos from long-term subcultured callus. Scale bar = 1.0 cm. (**B**) Stereomicroscope close-up of (**A**). Scale bar = 0.1 cm. (**C**) Number of mature somatic embryos from thawed cryopreserved callus and long-term subcultured callus. Data represent mean ± SD. Different lowercase letters above the bars indicate the differences among PEG treatments within the same embryogenic cell line according to ANOVA and Duncan’s test (*p* < 0.05).

**Figure 4 plants-14-01569-f004:**
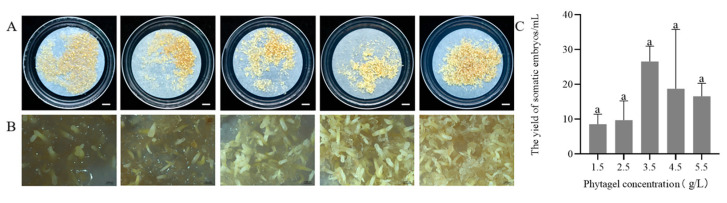
Effect of phytagel concentrations on somatic embryo maturation from thawed cryopreserved callus in *P. massoniana* (75-day culture). (**A**) Somatic embryos derived from thawed callus cultured on maturation media with different phytagel concentrations (left to right are 1.5, 2.5, 3.5, 4.5, and 5.5 g/L). Scale bar = 1.0 cm. (**B**) Stereomicroscope close-up view of (**A**). Scale bar = 0.1 cm. (**C**) Number of mature somatic embryos at different phytagel concentrations. Data represent mean ± SD. The same lowercase letters above the bars indicate no significant differences among phytagel treatments within the same embryogenic cell line according to ANOVA and Duncan’s test (*p* < 0.05).

**Figure 5 plants-14-01569-f005:**
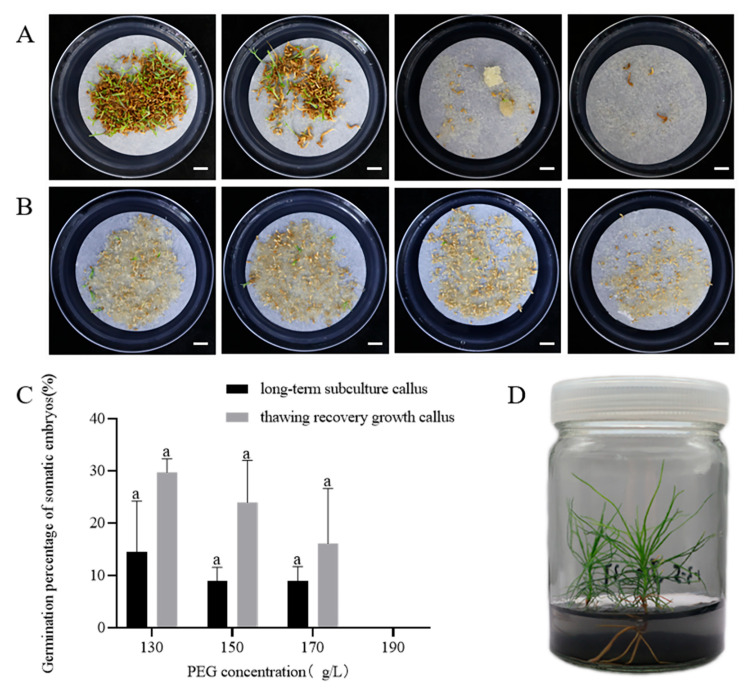
Effect of PEG concentrations on somatic embryo germination in *P. massoniana* (30-day culture). (**A**) Germinated somatic embryos derived from thawed cryopreserved callus cultured on media with different PEG concentrations (left to right are 130, 150, 170, 190 g/L). Scale bar = 1.0 cm. (**B**) Germinated somatic embryos derived from long-term subcultured callus under the same PEG conditions as (**A**). Scale bar = 1.0 cm. (**C**) Germination rate of somatic embryos under different PEG concentrations (130, 150, 170, 190 g/L). (**D**) Regenerated plantlets in sterile tissue-culture containers. Data represent mean ± SD. The same lowercase letters above the bars indicate no significant differences among PEG treatments within the same embryogenic cell line according to ANOVA and Duncan’s test (*p* < 0.05).

**Figure 6 plants-14-01569-f006:**
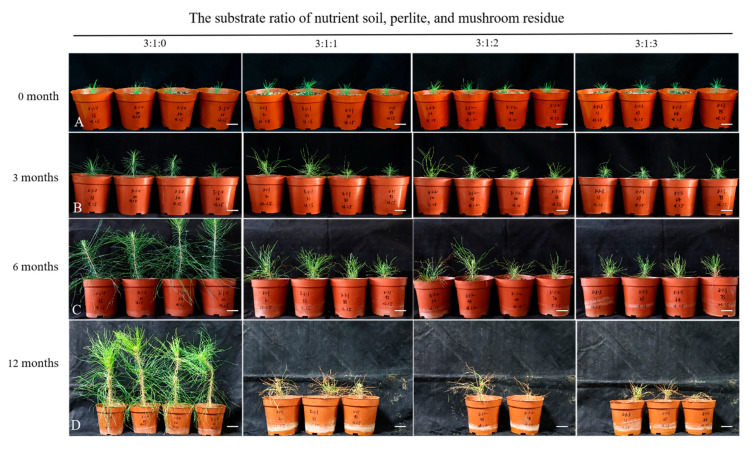
Growth performance of *P. massoniana* regenerated plantlets in different substrate mixtures over 12-month cultivation. (**A**–**D**) Plantlet development at (**A**) 0, (**B**) 3, (**C**) 6, and (**D**) 12 months after transplantation. Scale bar = 1.0 cm. Substrate ratios (nutrient soil: perlite: mushroom residue) from left to right: 3:1:0, 3:1:1, 3:1:2, 3:1:3.

**Table 1 plants-14-01569-t001:** Effects of different substrates on the growth of plantlets of *P. massoniana* (*p* < 0.05).

The Ratio of Nutrient Soil, Perlite, and Mushroom Residue	Plants Height (cm)				Plant Height Increment (cm)			Transplant Survival Rate (%)		
	0 Month	3 Months	6 Months	12 Months	3 Months	6 Months	12 Months	3 Months	6 Months	12 Months
3:1:0	2.4 ± 0.8 a	4.6 ± 1.6 a	15.0 ± 4.4 a	26.9 ± 5.9 a	2.3 ± 1.6 ab	12.6 ± 4.4 a	22.3 ± 5.4 a	56.5	56.5	56.5
3:1:1	2.5 ± 0.8 a	5.1 ± 2.1 a	7.3 ± 2.8 b	11.4 ± 3.7 b	2.6 ± 1.6 a	4.8 ± 2.8 b	5.0 ± 2.6 b	95.7	82.6	26.1
3:1:2	2.5 ± 0.8 a	4.7 ± 1.9 a	5.8 ± 1.9 bc	6.2 ± 1.5 bc	2.1 ± 1.4 ab	3.3 ± 1.9 bc	1.1 ± 0.6 b	95.7	78.3	21.7
3:1:3	2.5 ± 0.6 a	4.1 ± 1.2 a	4.8 ± 1.19 c	5.2 ± 0.9 c	1.6 ± 0.8 b	2.3 ± 1.2 c	1.2 ± 0.5 b	100	79.2	21.7

Note: Different lowercase letters indicate the differences among substrates treatments within the same embryogenic cell line according to ANOVA and Duncan’s test (*p* < 0.05).

## Data Availability

The original contributions presented in the study are included in the article. Further inquiries can be directed to the corresponding author.

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
