# Peer review of "Cryopreservation and Maturation Media Optimization for Enhanced Somatic Embryogenesis in Masson Pine (*Pinus massoniana*)"

_plants, 2025, doi:10.3390/plants14111569_

Round 1
Reviewer 1 Report
Comments and Suggestions for Authors
The manuscript provides a comprehensive and methodologically well-structured study on optimizing somatic embryogenesis in Pinus massoniana through cryopreservation and media optimization, with the ultimate goal of supporting disease-resistant breeding programs. The following suggestions are requested
Major concern
- Low Recovery Rate, Only one out of five cryopreserved ECs showed successful recovery (20%). Further optimization or explanation is needed.
- No Genetic Stability Assessment, The authors did not analyze genetic fidelity post-cryopreservation, which is crucial in SE-based propagation. Is it possible to include standard techniques like SSR markers, RAPD, or flow cytometry ? or at least discussed.
- Single Cell Line Reliance, All optimization and conclusions are based on one cryopreserved and one long-term subcultured cell line. Genotype variability is a known factor in SE success, and can findings be generalizable?
- Although figures provide means and standard deviations, statistical tests (such as ANOVA results, significance levels) are not consistently reported, especially in results like germination and survival (for intstance, Figure 5C, Table 1).
- Numerous grammatical errors and awkward phrasing throughout the manuscript detract from clarity and professionalism. A thorough language edit is strongly recommended.
- Without line numberting, it is very diffcult make coments line by line
Minor Comment
Excellent summary of findings. However, include statistical significance values if possible.
Use consistent units: “−196 °C” is fine, but later “3.5 g/L” should also include standard SI spacing.
“poses a major threat…” rewrite sentence; readability will increase
“SE is a promising tool…” Good justification, but condense background literature slightly.
Good summary of cryopreservation benefits, but lack of transition between paragraphs.
Add a sentence on why Pinus massoniana is a good model or case species.
The objectives are well structured.
A 20% recovery rate is too low; consider proposing improvements or addressing why.
Good histological comparison, but please clarify how many replicates were used.
Present statistical test results with p-values.
Define “columnar” in context of embryo shape, or provide figure cross-reference.
Waterlogging is vaguely mentioned; discuss how it was quantified or observed.
Clarify if germination tests were done in light or dark and under what temperature.
The survival rate plateau (56.5%) needs a biological explanation.
Table 1, Add statistical significance (for instance, superscripts for different groups).
Nicely organized and connects literature, though it occasionally reiterates results.
You acknowledge DMSO toxicity, please suggest testing alternatives in future work.
Too lengthy; consider summarizing ABA and PEG discussion into a single paragraph.
Good coverage of phytagel effects, but more clarity needed about gelling strength.
Acclimatization findings are well presented — suggest mentioning potential mycotoxin influence from mushroom residue.
Suggest a future direction to include more genotypes for validation.
Sufficient detail is provided. Please clarify:Incubation conditions (light intensity, humidity), number of replicates used for each treatment, whether experiments were repeated
Change “standard error” to “standard deviation” if figures use SD.
“These results provide… organized summarization.
Rephrase “Future research should assess…” to “Further validation using multiple genotypes is recommended…”.
Author Response
Please refer to the annex.

Reviewer 2 Report
Comments and Suggestions for Authors
The article " Cryopreservation and maturation media optimization for enhanced somatic embryogenesis in masson pine (Pinus massoniana)" is interesting and well written. This is a good, well-reasoned study devoted to the conservation of the Pinus massoniana Lamb. (masson pine) species. The article is certainly worthy of publication.
Minor comments:
- " Cryopreservation and maturation media optimization for enhanced somatic embryogenesis in masson pine (Pinus massoniana)" The title of the work does not reflect the content of the article. In addition to the issues of embryogenesis and cryopreservation, the authors study the composition of the soil for plant adaptation. It is necessary to think about changing the title of the manuscript.
- The introduction should justify the choice of ABA, PEG 8000 and Phytagel in the somatic embryogenesis protocol.
Results
- 1.2. Microstructure of P. massoniana embryogenic callus before and after cryopreservation
«proembryo genic masses (PEM) III and SE stages». Characterize these stages. They should be shown in Figures 1 or 2.
- 2.2.2. Effect of PEG 8000 concentrations on somatic embryo maturation
«embryos were mostly columnar» - what does this mean? Show this in Figure 3.
Methodology
5. 4.1. Cryopreservation of embryogenic callus of P. Massoniana
What is the composition of the proliferation medium? Under what conditions (temperature, humidity, lighting) were the calli cultivated and in what vessels (vessel volume, amount of nutrient medium per callus)?
6. 4.2. Effect of transplanting substrate on growth and survival of regeneration plantlet
«mushroom residue» - what is it, how was it obtained? What fungi were present?
Author Response
Please refer to the annex.

Reviewer 3 Report
Comments and Suggestions for Authors
The study titled “Cryopreservation and maturation media optimization for enhanced somatic embryogenesis in masson pine (Pinus massoniana)”, although I consider that this title does not correctly describe the entire study, I suggest a better alternative to describe the whole work.
Here are some suggestions, and I also made a few observations in the attached manuscript file:
In Figures 1 and 2, it would be beneficial to indicate the structures of the embryos with arrows (embryonic head and suspensor cells, split embryonic head, etc.).
The images in Figures 2, 3, and 4 are very small and difficult to distinguish. I suggest enlarging them and setting the graphs aside. The scale bars are tiny and not noticeable.
The Y-axis scale in graphs F and G in Figure 2 must have the same values for a better comparison; that is, the maximum value in both graphs must be 50.
In section 2.2.2, the authors mention that “embryos from long-term subcultured callus were predominantly columnar.” What stage is that? Generally, the recognized stages are globular, heart-shaped, torpedo, and cotyledon.
In section 2.2.3, the authors mention, “the callus exhibited severe waterlogging.” Do they refer to hyperhydration?
I suggest replacing Table 1 with a line graph to show the increase or decrease in growth of the regenerated seedlings over the 12 months.
In the Discussion section, the authors mention, "This result is consistent with previous reports in P. koraiensis (Gao et al. 2023) and P. pinaster (Marum et al. 2004)." They should elaborate on which study results coincide with this study's.
Mention in the discussion how and to what extent different Phytagel concentrations affect the medium's osmotic pressure values.
In Section 4. Materials and Methods: Please mention previous studies that established the growth, maturation, and germination media.
In Section 4.4, what type of agar was used? And why did Phytagel not continue to be used for germination and plantlet regeneration?
The paragraph in section 4.4.1 is confusing; I think it should be integrated into section 4.4. Somatic embryo germination and plantlet regeneration.
Section 4.5. Statistical analysis does not specify which data were analyzed or whether a homoscedasticity test was performed to verify the normality of variances. It also does not indicate which post hoc test was performed to identify significant treatment differences.

Author Response
Please refer to the annex.

Round 2
Reviewer 1 Report
Comments and Suggestions for Authors
Thank you for the revision
Author Response
Thank you very much for your positive feedback and for the thorough review of our revised manuscript. We appreciate the time and effort you have invested in evaluating our work. We have carefully considered all the comments and suggestions provided, and have made the necessary revisions to enhance the quality and clarity of our manuscript. We believe that these changes have significantly improved our work and addressed the concerns raised during the review process.